# Does Listener Gaze in Face-to-Face Interaction Follow the Entropy Rate Constancy Principle: An Empirical Study

**Yu Wang** and **Hendrik Buschmeier**
Digital Linguistics Lab, Department Linguistics
Faculty of Linguistics and Literary Studies
Bielefeld University, Bielefeld, Germany
{y.wang,hbuschme}@uni-bielefeld.de

## Abstract

It is generally assumed that language (written and spoken) follows the entropy rate constancy (ERC) principle, which states that the information density of a text is constant over time. Recently, this has also been found for nonverbal gestures used in monologue, but it is still unclear whether the ERC principle also applies to listeners' nonverbal signals. We focus on listeners' gaze behaviour extracted from video-recorded conversations and trained a transformer-based neural sequence model to process the gaze data of the dialogues and compute its information density. We also compute the information density of the corresponding speech using a pre-trained language model. Our results show (1) that listeners' gaze behaviour in dialogues roughly follows the ERC principle, as well as (2) a congruence between information density of speech and listeners' gaze behaviour.

## 1 Introduction

Human social interaction is intrinsically multimodal (Stivers and Sidnell, 2005). Face-to-face communication as a multimodal process includes verbal information as well as non-verbal cues, such as gaze, head movements, and speech-accompanying manual gestures from both interlocutors. Previous studies have demonstrated that non-verbal behaviours is rich in communicative functions (Wagner et al., 2014; Holler and Levinson, 2019).

In this paper, we look at gaze behaviour of listeners in video-recorded face-to-face-interaction, specifically, explanation dialogues in which a board game is explained by one interlocutor (the 'explainer') to another (the 'explainee'; Türk et al., 2023). We apply information theoretical measures to the gaze behaviour of the explainees as well as the corresponding utterances of both interlocutors and aim at answering the following questions: (i) How informative is listener gaze from an interactional perspective and does it follow the 'entropy rate constancy' principle (ERC; Genzel and Charniak,

2002)? (ii) Is there a correlation between verbal information and listener gaze, in terms of local entropy, in dialogue (similar to recent findings on manual gesture in monologue; Xu et al., 2022)?

## 2 Related work

### 2.1 Communicative functions of gaze

In human interaction, gaze is an important and multifunctional nonverbal signal with functions such as indicating attention, allocating space, and eliciting and monitoring feedback (Kendon, 1967; Duncan, 1975; Harness Goodwin and Goodwin, 1986). According to Brône et al. (2017), these functions follow two important roles of gaze: a participation role and a regulation role. They also emphasise the importance of gaze for turn management. During dialogue interaction, listeners' continuous gaze towards the speaker signals attention and engagement (Kendon, 1967; Rossano, 2012).

From the technical side, the importance of gaze has been studied and integrated in recovering meaning in interactional dialogues (Alaçam et al., 2021), facilitation grounding in interaction with embodied agents (Nakano et al., 2003) or in human-robot interaction (Skantze et al., 2014).

### 2.2 Entropy-rate constancy

Information theory (Shannon, 1948) is a mathematical framework that has proved useful for linguistic analysis as it can explain aspects of language use. Genzel and Charniak (2002, 2003), for example, propose the 'Entropy Rate Constancy' (ERC) principle, which states that entropy is constant over the length of a written text or other use of language.

The claim by Genzel and Charniak (2002) is as follows: Let $H(X_i|C_i, L_i)$ denote the conditional entropy of the word $X_i$, where $L_i = X_{i-n+1}, \ldots, X_{i-1}$ is the local $n$-gram context, and $C_i = X_0, X_1, \ldots, X_{i-1}$ the context which contains all of the words preceding the word $X_i$. The condi-

tional entropy of the word $X_i$ can then be decomposed as:

$$\underbrace{H(X_i|C_i, L_i)}_{\text{global entropy}} = \underbrace{H(X_i|L_i)}_{\text{local entropy}} - \underbrace{I(X_i; C_i, L_i)}_{\text{mutual information}}$$

The assumption of ERC is that $H(X_i|C_i, L_i)$ is constant. Given the fact that $I(X_i; C_i, L_i)$ – as the mutual information between $X_i$ under its local context $L_i$ and its global context $C_i$ – increases because the global context increases, the local entropy $H(X_i|L_i)$ would have to increase in order for $H(X_i|C_i, L_i)$ to remain constant.

A similar theory, the 'Uniform Information Density' (UID) hypothesis, states that speakers tend to distribute information uniformly throughout an utterance (Jaeger, 2010). In UID, the information of a linguistic unit $y$, defined by its surprisal, is its negative log probability $s(y) = -\log_{p_\ell}(y)$, where $p_\ell$ is the underlying probability distribution of $y$. Since the true probability distribution of $y$ is not available, a language model with learned parameters is usually used to approximate the surprisal value of the corresponding linguistic unit (Smith and Levy, 2013; Goodkind and Bicknell, 2018; Wilcox et al., 2020). The linguistic unit $y$ as a large sequence (e.g., an utterance or a text), can be further divided into a sequence of smaller units: $\langle y_1, \ldots, y_n \rangle$, where $y_n \in \vartheta$ and $\vartheta$ is the set of vocabulary. The surprisal of the current linguistic unit $y_n$ is then expressed as the conditional negative log probability given its previous context: $s(y_n) = -\log_{p_\ell}(y_n|y_{<n})$. According to the UID hypothesis, drastic variations in the per-unit information density of an utterance can place a heavy processing burden on the listener, and thus make communication more difficult. The evenly distributed information density of speech on the other hand, promotes 'rational' (Xu and Reitter, 2018) communication.

More recent studies have applied the ERC principle and/or UID hypothesis to spoken dialogue (Xu and Reitter, 2017, 2018), task oriented dialogue (Giulianelli et al., 2021), as well as non-verbal signals (specifically manual gesture) in monological speech (Xu et al., 2022).

## 3 Hypothesis

In this paper, we investigate the hypothesis that, during face-to-face interaction, listener gaze, being an important non-verbal communication mechanism, also conforms to the ERC principle.

## 4 Methods

### 4.1 Data collection

The dialogue interactions on which this study is based are explanations of board games (Türk et al., 2023). The explainer, who is familiar with the game, explains it to the explainee, who is unfamiliar with the game (see Figure 1). There are three reasons why we focused on task-oriented dialogue, namely the game explanation scenario, in our study: (i) Based on the theory of topic shifts (Ng and Bradac, 1993; Linell, 1998), we know that some interlocutors play a more active role, controlling the dialogue and introducing new topics, while the others have a more passive role and follow these topic shifts. Therefore, we identify explanations as common and representative of daily dialogue (there is always a more dominant speaker – the explainer – and a more passive listener). Furthermore, we assume that the passive listeners instead use more non-verbal signals (e.g. gaze, facial expression) than verbal signals to give feedback to the dominant speakers (Buschmeier and Kopp, 2018), which potentially provides different gaze data for this study. (ii) By focusing on one type of dialogue (explanations) and additionally one topic (a specific game), we assume that the dialogue contents are quite static at the lexical and semantic level and can be considered as invariant in our experiment (and effects on the entropy rate of gaze are not by the domain). (iii) Based on this, we can, in principle, even look at the behaviour of individual speakers and whether their speech behaviour follows the ERC principle.

The interactions are divided into two phases: In the first phase, the game is explained without the game material being present. In a second phase, the game is put on the table and the two participants start playing (but usually the explanation continues at least while the game is being set up). All participants speak German. The interaction can therefore be considered a task-oriented dialogue. For this study we use the videos of 58 interactions from the corpus and extract the explanation part (first phase). The explanations vary in length from 2:12 to 17:36 min (mean length 7:04 min, standard deviation 3:15 min).

### 4.2 Preparing gaze sequences

For each dialogue, we extract the explainee's gaze information using the 'Openface' framework (Baltrusaitis et al., 2018) and create a 'gaze sequence' that is used in the neural sequential model (see

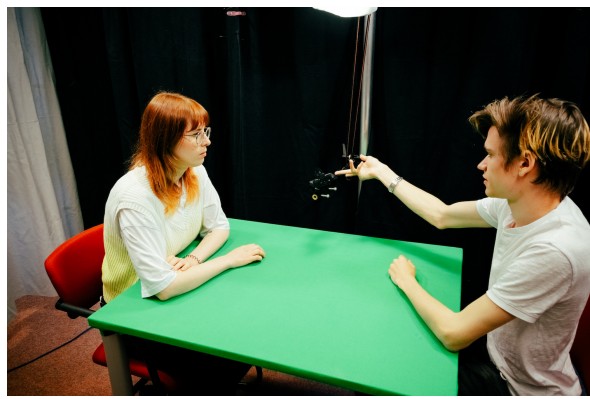

Figure 1: Scene from the data collections. The participant on the left (the explainer) explains the board game to the participant on the right (the explainee). The explainee's eye gaze is captured with a camera behind the explainer.

|  | left of DR | DR | right of DR |
|---|---|---|---|
| Above DR | 7 | 8 | 9 |
| DR | 4 | 5 | 6 |
| Below DR | 1 | 2 | 3 |

Table 1: Label decisions criterion for direction value representation of eye gaze (DR: dense region).

|  | RE forward | DR | RE backward |
|---|---|---|---|
| LE forward | 9 | 8 | 7 |
| DR | 6 | 5 | 4 |
| LE backward | 3 | 2 | 1 |

Table 2: Label decisions criterion for vector representation of eye gaze (RE: right eye; LE: left eye).

Section 4.4). Openface generates two types of gaze features (i) *gaze vectors* in world coordinates (three dimensions for the left and right eye each), and (ii) *gaze direction values* in radians averaged from both eyes (two dimensions). We integrate both representations because gaze direction is easier to use, but does not contain depth information, which is potentially relevant as listeners regularly change their posture and head orientation during the interaction.

We cluster the two gaze features using the DBSCAN algorithm (Ester et al., 1996) to find the spatial distribution of gaze and identify its 'dense region' (Tran et al., 2020), both horizontally and vertically. This dense region typically represents the target which explainees gaze at most of the time during an interaction. After the dense region has been detected, we use a $3 \times 3$ grid-based labelling scheme (inspired by Xu et al., 2022) and label the gaze points inside the dense region with '5'. Depending on whether the gaze direction is horizontally or vertically away from the dense region, eight other number-based labels are used for the gaze points outside the dense region (see Table 1). A similar approach is used for the depth-component selected from the eye gaze vectors. DBSCAN-clustering is used to find the dense region where the gaze of the left eye and right eye are located in the depth dimension. The eye gaze vector inside the dense region is again given the label '5'. Based on how close the left eye vector or right eye vector is to the dense region, eight different labels are used (see Table 2).

Given the label $x_d \in [1, 9]$ for the gaze direction value and the label $x_v \in [1, 9]$ for the gaze vector, a combined label $y$, that represents the gaze informa-

tion, is generated as $y = (x_d - 1) \cdot 9 + x_v$ ($y \in [1, 81]$ representing the set of possible eye gaze labels).

## 4.3 Automatic speech recognition for the interaction

Transcriptions of the dialogues were created automatically using 'Whisper' (Radford et al., 2022), which creates speech segments with a start and an end time. In order to calculate word timings, we approximated word onsets by calculating the duration of each speech segment, dividing it by its length in words, and approximating word duration (assuming, for this study, that words have uniform length). Eye gaze labels are then aligned with words based on video frame rate (50 fps). Figure 2 shows an example of this alignment. After pre-processing, we concatenate all of the utterance-aligned gaze sequences and use them as training and test data for the neural sequential model.

## 4.4 Processing gaze sequences

Analogous to the processing the information density of linguistic units according in the UID hypothesis (see section 2.2), we consider eye movements as

| Das | Problem | ist | wir | haben | nur | ein | billiges | U-Boot | wo |
|---|---|---|---|---|---|---|---|---|---|
| 42 | 42 |  | 42 | 42 | 42 | 42 | 42 | 42 | 42 |

| wenig | Sauerstoff | drin | ist | und | wir | müssen | uns | den | teilen |
|---|---|---|---|---|---|---|---|---|---|
| 42 | 42 |  | 42 | 42 | 42 | 41 | 45 | 42 | 45 | 45 |

| Wir | sind | also | eigentlich | Gegner | wir | spielen | gegeneinander |
|---|---|---|---|---|---|---|---|
| 45 | 45 | 45 | 45 |  | 45 | 45 | 44 | 44 |

| aber | wir | nutzen | den | gleichen | Sauerstoff |
|---|---|---|---|---|---|
| 41 | 41 | 41 | 41 | 41 | 41 |

Figure 2: Example of the alignment of speech segments (grey) and gaze label sequences (white).

sequences of gaze labels and need to estimate its underlying probability distribution. We approach this by training an autoregressive model, more specifically a Transformer model (Vaswani et al., 2017), which we have chosen since it has a strong psychometric predictive power compared to LSTM-RNNs models(Wilcox et al., 2020). To compute the local entropy of a gaze sequence, we first calculate its negative log probability:

$$\text{NLL}(e_1, e_2, \ldots, e_T)$$
$$= -\big( \log P(e_1) + \sum_{i=2}^{T} \log P(e_i | e_1, \ldots, e_{i-1}) \big)$$

where $T$ is the maximum index of a given eye movement sequence and $e_i \in [1, 81]$. The local entropy $H(e_1, e_2, \ldots e_i)$ of the gaze sequence is then the exponential of NLL (perplexity). The learning task is thus to predict the next gaze label $e_i$ based on the preceding sequence $\langle e_1, \ldots, e_{i-1} \rangle$ and minimise its negative log probability NLL.

## 4.5 Processing dialogue data

To compare the local entropy of the gaze sequences with the local entropy of the corresponding speech segments, we compute the latter using a pre-trained language model (specifically dbmdz/german-gpt2; Schweter, 2020). For the computation we use whole dialogues, i.e., both explainer and explainee utterances, as both contribute to the dialogue and explainee utterances are conditioned on what the explainer has said before and vice versa. Some of the utterances contain only backchannels (such as "yes", "uh-huh", or "okay"; Yngve, 1970), which play an important role in the dialogue as linguistic 'feedback' to previous utterances (Allwood et al., 1992; Clark, 1996). In our data, backchannels are usually produced by explainees and we do not exclude them as semantically irrelevant words.

## 5 Results and discussion

Figure 3 shows the combined eye gaze sequences from all 58 videos. The x-axes represent the dialogue position of the speech segments (with each dialogue position corresponding to about 7s of speech) and the y-axes represent the local entropy of the gaze sequences. Figure 3A, shows a generally rising trend of the local entropy, Figure 3B shows a replot of the data without the effect from entropy spikes shown in 3A (which we consider as outlier

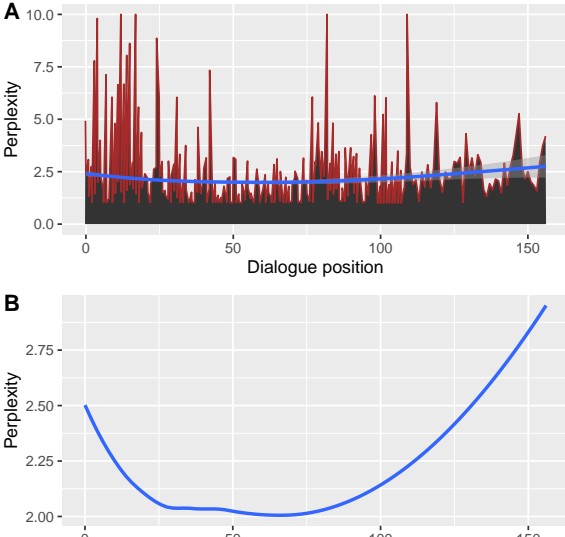

Figure 3: Change of the perplexity (local entropy) of eye gaze during the dialogues. A shows the general trend, B shows the normalised trend without the effect from the entropy spikes.

values), and yields a globally rising trend, but also a decreasing trend for the first 75 dialogue positions.

The 58 explanations vary in length (see Section 4.1), so plotting them together introduces some bias into the visualisation. We have therefore divided the dataset into four groups (based on the total length of the explanations) and plotted the local entropy of the gaze sequences separately in Figure 4. Besides the common rising trend in all of four sub-plots Figure 4A–D, they also share a decreasing tendency at the beginning, resulting in (roughly) convex shapes. One explanation for this phenomenon may be that, at the beginning of the interaction, explainees focus on the explanation of the game and direct their gaze (and attention) mainly to the explainer, thus signalling their participation role (Brône et al., 2017). As a result, there is little variation of gaze labels. However, as the explanation progresses, the explainees' cognitive load may increase, a known cause of gaze aversive behaviour (Morency et al., 2006; Doherty-Sneddon and Phelps, 2005). Another explanation for the increasing trend could be that explainers are likely to provide non-verbal feedback to explainers (e.g. head nodding or other head movements; Heylen, 2006), signalling understanding etc., which could also lead to changes in gaze behaviour. In any case, the result is a higher diversity of gaze labels and thus an increase in local entropy.

We chose a board game explanation so that the

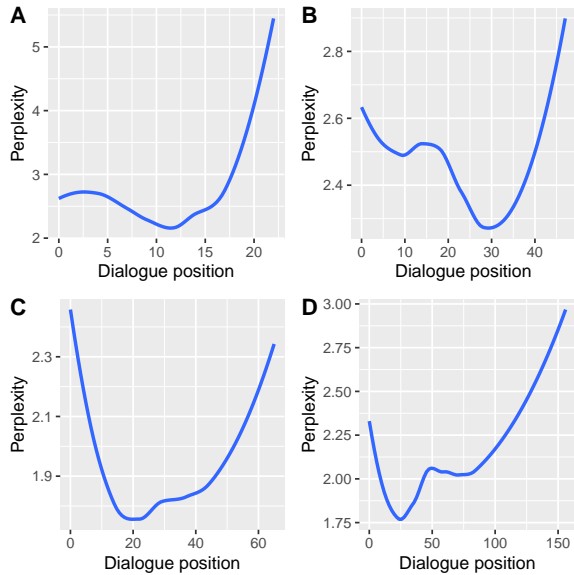

Figure 4: General trends in perplexity (local entropy) of explainees' eye gaze during explanations, grouped by length of dialogues (in dialogue positions): A: 4 dialogues shorter than 25; B: 29 dialogues between 25 to 49; C: 10 dialogues between 50 to 74; D: 15 dialogues longer than 74.

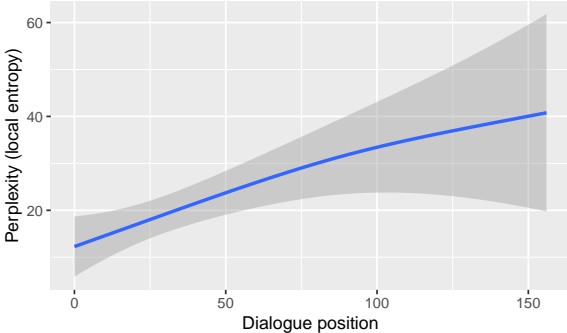

Figure 5: Change of perplexity(local entropy) of the speech segments during the dialogue.The shaded area is the bootstrapped 95% confidence intervals

linguistic content of the dialogue is limited. This gives us a good opportunity to look at whether different individuals organise their speech in a 'rational' way (following the ERC principle; Xu and Reitter, 2018). According to Figure 5, it turns out that, given the same topic (the explanation of the board game), different explainers organise the explanation in a rational way, as evidenced by the increasing trend of the local entropy. Moreover, this increasing trend of local entropy for both the speech segments across dialogues and their corresponding gaze sequences indicates a potential congruence between the information density of speech and the gaze behaviour of explainees/listeners.

## 6 Conclusions

This study attempts to find out whether (i) gaze as an important non-verbal communication signal follows the entropy rate constancy principle, and whether (ii) the information density of listeners' gaze behaviour correlates with that of the dialogue content. We recorded interaction videos, trained a transformer model, and used a pre-trained language model to approximate the information density of listeners' gaze and dialogue content.

We find that listeners' gaze roughly follows the ERC principle, which can be taken as further evidence that non-verbal communication generally follows the ERC principle (at least to some extent; Xu et al., 2022) – although the result is inconclusive in that the information density of listeners' gaze fluctuates along the interaction. A congruence between dialogues content and listeners' gaze can be roughly confirmed, as the local entropy of listeners' gaze and the local entropy of speech both show an increasing tendency. The fluctuation of the entropy rate value (Figure 3A as well as Figure 4) indicates that the property of non-verbal communication cannot simply be explained by ERC principles.

As a next step, we plan to look more closely at listeners' eye gaze and further analyse whether sudden changes in the local entropy of gaze behaviour can be aligned with changes in the local entropy of speech, and if so, under what kinds of context. We also plan to perform further linguistic analysis on the entropy spikes, looking at dialogue acts and calculating the syntactic complexity of the corresponding dialogue positions. Possible future research includes simplifying or compressing the dialogue context based on changes in the entropy values of the listener's gaze.

**Supplementary materials** Code and data of analyses is available in the following data publication: https://doi.org/10.6084/m9.figshare.24408073

**Acknowledgements** This work was supported by the German Research Foundation (DFG) in the Collaborative Research Center TRR 318/1 2021 'Constructing Explainability' (438445824).

We would also like to acknowledge Olcay Türk, Stefan Lazarov, and our student assistants for the data collection, Özge Alaçam for feedback on the manuscript, and the anonymous EMNLP reviewers for their valuable comments.

## Limitations

One limitation of the study is that the dialogues are not very diverse in content and activity. They are all explanations of a specific board game.

A second limitation is that because we rely on video-based gaze information collection rather than dedicated eye-tracking hardware, the gaze data derived from Openface is sometimes sub-optimal – the camera setup was not optimised for gaze and face tracking, as the camera shots were over the shoulder and thus slightly elevated, rather than at eye level – so the data may contain noise. This compromise was necessary to meet multiple analysis objectives and also to avoid disrupting the naturalness of the conversational setting (e.g., by using wearable eye-trackers).

A third limitation concerns the alignment of utterances with gaze labels. Since German words can be very long (e.g. 'Tiefseeabenteuer', the name of the board game), the simplified assumption that all words have the same duration and align with the same number of gaze labels will cause some noise.

## Ethics statement

The data collected for this study is for research purposes only and no commercial use is allowed. The participants recruited are mainly university students. Participants gave informed consent for their participation in the study and the use of their data, and were paid 10 euros per hour. The study was approved by the university's internal ethics and data protection review boards.

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

## A Fragment of a game explanation

Figure 6 shows an example fragment of a game explanation. The timestamped speech segments were automatically recognised from the audio of the interaction videos using 'Whisper' (Radford et al., 2022). Whisper's word error rate (WER) for German is given as 4.5% (see `https://github.com/openai/whisper/blob/main/README.md`). The transcripts were not corrected.

## B Hyperparameters selection for DBSCAN clustering

For the DBSCAN algorithm, we set up a criterion that only if a point is surrounded by at least ten samples, it can be considered a core point for a cluster. A for-loop is used to find an optimal epsilon value required by the DBSCAN algorithm. The epsilon value is in the range $[0.01, 0.1]$. We wanted

| | |
|---|---|
| 00:00.000 → 00:04.820 | *Also grundsätzlich das Spiel heißt Tiefseeabenteuer und es geht quasi darum, du bist ein Taucher* |
| | So basically the game is called Deep Sea Adventure and it's basically about you being a diver |
| 00:04.820 → 00:05.820 | *in einem U-Boot* |
| | in a submarine. |
| 00:05.820 → 00:09.280 | *Du musst möglichst lange raustauchen, um dir die Schätze anzusammeln.* |
| | You have to dive out as long as possible to collect the treasures. |
| 00:09.280 → 00:12.800 | *Wer am Ende mehr Schätze hat, hat gewonnen, das ist so der Grundaufbau.* |
| | Whoever has more treasures at the end wins, that's the basic structure. |
| 00:12.800 → 00:18.400 | *Du musst dir vorstellen, du hast hier das U-Boot, da starten wir auch drin und da hast* |
| | You have to imagine that you have the submarine here, we start in there and there you have |
| 00:18.400 → 00:25.480 | *du dann quasi 25 Felder, bei 25 bis 0, das ist im Ende unsere Sauerstoffanzeige und die* |
| | you then basically 25 fields, at 25 to 0, that's ultimately our oxygen display and these |
| 00:25.480 → 00:28.240 | *teilen wir uns, das wird gleich noch wichtig, die hat nicht jeder einzeln, wir teilen uns* |
| | we'll share, that'll be important in a moment, not everyone has them individually, we'll share |
| 00:28.240 → 00:31.720 | *zusammen einen Sauerstofftank und dann haben wir gleich ganz viele kleine Plättchen, da* |
| | together an oxygen tank and then we have a lot of small plates right there |
| 00:31.720 → 00:37.680 | *sind Punkte drauf von 1 bis 4, die geringsten 4, die höchsten werden dann eben von 1 bis* |
| | are points on it from 1 to 4, the lowest 4, the highest are then just from 1 to |
| 00:37.680 → 00:43.240 | *4 an einem U-Boot entlang ausgelegt, dass quasi ein Weg entsteht und dann haben wir* |
| | 4 laid out along a submarine so that a path is created and then we have it |
| 00:43.240 → … | … |

Figure 6: Example ASR transcript of a fragment of an explanation dialogue (with English translations).

to find an epsilon value that ensures that the ratio between the second most frequent cluster label and the most frequent cluster label is just below 15%, so that the cluster with the most frequent label is considered to be the region where an explainee's gaze is mostly located.

## C   Hyperparameter selection for the training procedure

For the neural sequence model (Transformer) the batch size is 35, the input size is set to 25, the hidden layer size is Data is divided into 80% for training and 20% for testing. The initial learning rate is 0.05. Both training and testing data sets are used to compute the local entropy.

To compute the local entropy for the speech segments, we use the pre-trained language model `dbmdz/german-gpt2` (Schweter, 2020), the input size is set to 25, which is consistent with the gaze sequence input size for the neural sequence model.