# OpenReview forum: "Does Listener Gaze in Face-to-Face Interaction Follow the Entropy Rate Constancy Principle: An Empirical Study"
_EMNLP/2023/Conference — EMNLP 2023 Findings_

### Official Review · Reviewer_V3pQ · 2023-07-28

**Soundness:** 2

**Excitement:**

3: Ambivalent: It has merits (e.g., it reports state-of-the-art results, the idea is nice), but there are key weaknesses (e.g., it describes incremental work), and it can significantly benefit from another round of revision. However, I won't object to accepting it if my co-reviewers champion it.

**Paper Topic And Main Contributions:**

This short paper investigates whether the entropy of the listener's gaze in face-to-face conversations follows the law of entropy rate constancy (ERC).
The current study focuses on scenes where participants describe board games without using game equipment.
Gaze information is extracted using OpenFace, and then the analysis confirms that Perplexity increases with the length of the conversation.

**Reasons To Accept:**

- While linguistic features on ERC have been studied before, this research is the first attempt to investigate gaze behavior.
- The use of OpenFace for gaze information extraction may not be perfect, but the methodology is carefully explained, ensuring reproducibility.

**Reasons To Reject:**

- The current study only uses one type of dialogue, limiting the generalizability of the experimental results.
- The criteria for excluding outliers in the analysis appear arbitrary.
- Although the experimental results are intriguing, the Introduction fails to adequately explain their practical significance.

**Reproducibility:**

4: Could mostly reproduce the results, but there may be some variation because of sample variance or minor variations in their interpretation of the protocol or method.

**Reviewer Confidence:**

2: Willing to defend my evaluation, but it is fairly likely that I missed some details, didn't understand some central points, or can't be sure about the novelty of the work.

---

> ### Author Rebuttal · Authors · 2023-08-28
>
> Thank you very much for reading our paper and giving us your valuable feedback. We consider your three concerns as follows:
>
> 1. (limited generalisability of the experiment results cause by one type of dialogue): There are three reasons, why we focussed only on task-oriented dialogue in this study. (i) Based on the theory of topic shifts (e.g., Ng and Bradac, 1993; Linell, 1998) we know that some interlocutors play a more active role controlling the dialogue and introducing new topics, while the others has a more passive role and follows those topic shifts. Based on this we identified explanation as common and representative of daily dialogue (there is always a more dominant speaker – the explainer – and a more passive listener). We thus think that our chosen domain is quite representative for dialogues in general. (ii) Focussing on one type of dialogue (explanations) and additionally one topic (a specific game) we assume that the content is quite static in the dataset (from the lexical and semantic levels), and can be considered invariant in our experiment. With the dialogue contents as the invariant, the results on entropy rate of the gaze will be more convincing. (iii) Based on this, we can even look at the behaviour of individual speakers and whether their speech behaviour follows the ERC theory (see figure 5). More generally we also see that it would be interesting to look at a more diverse set of dialogues (e.g., more casual conversations, different board games, …) and would like to see our result being replicated with other data.
>
> 2. (outlier exclusion): We would like to clarify that our criteria for excluding outliers are not arbitrary but based on empirical observation. Based on figure 3A, we found that in some limited dialogue positions, the perplexity jumps over 3.75 and falls back, which means that perplexity is usually below 3.75. Based on this point, we set up a threshold in our visualisation to exclude the effect of these spikes, which helps to generate the figure 3B where an increasing trend is more obvious. Thanks for pointing this out, we would explain it in more detail in a revised version.
> The perplexity-spike is a question we would like to answer, with more in-depth linguistic analysis,  in future work.
>
> 3. (introduction fails to adequately explain their practical significance): Our contribution is a theoretical one answering the two questions posed in the introduction (is there a relation between verbal information and listeners’ gaze; how informative is listener gaze). Our experiment shows a congruence between explanation (verbal information) and listener gaze, and that gaze behaviour roughly follows the ERC principle. On a more practical note, we think that these theoretical insights could serve as a basis for NLP applications incorporating gaze behaviour (mentioned briefly in the conclusion). We will make this more explicit.
>
> ## References
>
> * Linell (1998). Approaching dialogue: Talk, interaction and contexts in dialogical perspectives. John Benjamins.
> * Ng and Bradac (1993). Power in language: Verbal communication and social influence. Sage.

---

### Official Review · Reviewer_Ck9q · 2023-08-04

**Soundness:** 4

**Excitement:**

4: Strong: This paper deepens the understanding of some phenomenon or lowers the barriers to an existing research direction.

**Paper Topic And Main Contributions:**

This paper builds a computational pipeline to analyze the entropy of eye-gaze patterns during natural interlocutor dialogue and correlate this measure with entropy estimated from the time-aligned speech. This is a novel methods and while the results' consistency (or potential inconsistency) with entropy rate constancy is slightly murky, I think the methodological and paradigm contributes make this an interesting short paper for EMNLP

**Reasons To Accept:**

The paper is well-written and present a potentially interesting novel pipeline to link eye-gaze data during natural conversation with occurring speech.

Results are somewhat inconclusive, but the methods are themselves compelling and I think this is a good fit for a short paper of this type.

**Reasons To Reject:**

No reasons to reject.

**Reproducibility:**

4: Could mostly reproduce the results, but there may be some variation because of sample variance or minor variations in their interpretation of the protocol or method.

**Reviewer Confidence:**

4: Quite sure. I tried to check the important points carefully. It's unlikely, though conceivable, that I missed something that should affect my ratings.

---

> ### Author Rebuttal · Authors · 2023-08-28
>
> Thank you for your positive review of our paper. We are still cautious about our current results and will continue our ongoing work to reinforce our hypothesis on ERC in multimodal interactive communication (i.e., face-to-face dialogue) more generally.

---

### Official Review · Reviewer_H3tb · 2023-08-11

**Soundness:** 3

**Excitement:**

3: Ambivalent: It has merits (e.g., it reports state-of-the-art results, the idea is nice), but there are key weaknesses (e.g., it describes incremental work), and it can significantly benefit from another round of revision. However, I won't object to accepting it if my co-reviewers champion it.

**Paper Topic And Main Contributions:**

This paper studies whether the adherence of eye-gaze features to the ‘entropy rate constancy’ principle
(ERC). Toward that end, the authors conducted their study in the context of a board game explanation. The listeners are unfamiliar with the game and learn the game rules as an expert describes the game.
The empirical results demonstrate that listeners' eye gaze features adhere to the ERC principle of the experts' dialogue features.

**Reasons To Accept:**

Alignment study between non-verbal (eye gaze) and verbal (speech)

**Reasons To Reject:**

Dialogue position and gaze sequence alignment require more in-depth analysis

Specifically, Figure 4 demonstrates dialogue_positions and the change of perplexity of listeners' eye gaze.
A better analysis would be the linguistic features of dialogues and the change of perplexity of listeners' eye gaze.

**Reproducibility:**

4: Could mostly reproduce the results, but there may be some variation because of sample variance or minor variations in their interpretation of the protocol or method.

**Reviewer Confidence:**

3: Pretty sure, but there's a chance I missed something. Although I have a good feel for this area in general, I did not carefully check the paper's details, e.g., the math, experimental design, or novelty.

---

> ### Author Rebuttal · Authors · 2023-08-28
>
> Thank you for your review. In future work, we indeed plan to use more in-depth linguistic analyses in order to investigate why there are sudden increase/decrease of perplexity (see our conclusion). We plan to analyse it from two aspects: syntactic complexity and dialogue acts.
>
> Concerning the "reproducibility issue": We will release the code of our analyses alongside the paper, enabling reproduction on other data. We cannot release the multimodal data due to privacy reasons (GDPR) and limitations in the consent of the participants of our study.

---

### Meta-Review · Area_Chair_SiX8 · 2023-09-20

**Recommendation:** 3

**Metareview:**

The paper explores whether the listener’s non-verbal signal – mainly gaze, follows entropy rate constancy in dyadic conversation. Moreover, the paper also investigates if there is any alignment between the information density of speech content and the listener’s gaze behavior.
The study is inspired by Xu et al.'s 2022 study, which found that ERC applies to non-verbal gestures in monologues, this paper extends to test the principle to the listener's perspective in conversations.

While the reviews are mixed, most reviewers concur that the results presented are inconclusive. The paper could benefit from more comprehensive explanations and deeper analysis, especially concerning dialogue positions and gaze sequences. Despite being a short paper, the paper should aim for self-contained explanations, detailing the models used rather than solely referencing other works. Additionally, the paper lacks adequate justification for certain design and experimental choices, such as the removal of outliers, and should clarify the reliability and generalizability of the results (e.g., in Fig 4). The authors have acknowledged some of these issues and committed to addressing them in future iterations of the paper.

Given that the paper's research findings and methodologies hold some interest to the community, we recommend that the authors refine their work for the next version, taking the reviewers' feedback into account.

---

### Decision · Program_Chairs · 2023-10-07

**Decision:**

Accept-Findings

**Comment:**

The paper explores whether the listener’s non-verbal signal – mainly gaze, follows entropy rate constancy in dyadic conversation. Moreover, the paper also investigates if there is any alignment between the information density of speech content and the listener’s gaze behavior.
The study is inspired by Xu et al.'s 2022 study, which found that ERC applies to non-verbal gestures in monologues, this paper extends to test the principle to the listener's perspective in conversations.

While the reviews are mixed, most reviewers concur that the results presented are inconclusive. The paper could benefit from more comprehensive explanations and deeper analysis, especially concerning dialogue positions and gaze sequences. Despite being a short paper, the paper should aim for self-contained explanations, detailing the models used rather than solely referencing other works. Additionally, the paper lacks adequate justification for certain design and experimental choices, such as the removal of outliers, and should clarify the reliability and generalizability of the results (e.g., in Fig 4). The authors have acknowledged some of these issues and committed to addressing them in future iterations of the paper.

Given that the paper's research findings and methodologies hold some interest to the community, we recommend that the authors refine their work for the next version, taking the reviewers' feedback into account.